# Using A Soft Conformable Foot Sensor to Measure Changes in Foot Strike Angle During Running

**DOI:** 10.3390/sports7080184

**Published:** 2019-07-29

**Authors:** Herman van Werkhoven, Kathryn A. Farina, Mark H. Langley

**Affiliations:** 1Department of Health and Exercise Science, Appalachian State University, Boone, NC 28608, USA; 2Department of Human Physiology, University of Oregon, Eugene, OR 97403, USA

**Keywords:** foot strike pattern, running, sensor, inertial measurement unit

## Abstract

The potential association between running foot strike analysis and performance and injury metrics has created the need for reliable methods to quantify foot strike pattern outside the laboratory. Small, wireless inertial measurement units (IMUs) allow for unrestricted movement of the participants. Current IMU methods to measure foot strike pattern places small, rigid accelerometers and/or gyroscopes on the heel cap or on the instep of the shoe. The purpose of this study was to validate a thin, conformable IMU sensor placed directly on the dorsal foot surface to determine foot strike angles and pattern. Participants (n = 12) ran on a treadmill with different foot strike patterns while videography and sensor data were captured. Sensor measures were compared against traditional 2D video analysis techniques and the results showed that the sensor was able to accurately (92.2% success) distinguish between rearfoot and non-rearfoot foot strikes using an angular velocity cut-off value of 0°/s. There was also a strong and significant correlation between sensor determined foot strike angle and foot strike angle determined from videography analysis (*r* = 0.868, *p* < 0.001), although linear regression analysis showed that the sensor underestimated the foot strike angle. Conformable sensors with the ability to attach directly to the human skin could improve the tracking of human dynamics and should be further explored.

## 1. Introduction

Interest in running foot strike pattern (FSP) analysis has increased due to the potential association with reduced injury risk, better running economy, and improved performance [1,2,3]. Traditionally, FSP has been quantified using a variety of methods including two-dimensional (2D) video analysis, three-dimensional (3D) video analysis, center of pressure, and force plate readings [4,5,6,7]. Although these methods have been shown to be reliable tools to classify FSP, the ability to evaluate FSP in more ecologically valid settings is of interest. For example, it is known that treadmill running is not the same as over ground running [8]. Furthermore, even though videography allows for data collection outside the traditional laboratory setting, videography still requires a specific area of focus to accurately detect FSPs. Using these traditional methods, it is not possible, for example, to detect FSP during an entire 800 m middle distance race on a track, or during an entire road marathon event. Therefore, it is important to find practical and effective ways to evaluate FSP in any location where runners typically run.

The use of inertial measurement units (IMUs) has become a popular way to collect data in the field. The use of these systems in determining gait and FSP allows for unrestricted movement of the participants due to the IMUs’ small size and wireless capabilities. Various researchers have used accelerometers for determining changes in gait during walking and running [9,10,11,12,13,14,15]. Boutaayamou et al. (2015) [10] validated the use of two foot mounted accelerometers, one on the heel and one above the proximal end of the big toe on the shoe, in detecting heel-strike, toe-strike, heel-off, and toe-off during running gait against the conventional 3D analysis system. The authors found that using wireless accelerometers applied to the right and left foot could accurately and precisely detect these events in the gait cycle. Similarly, Giandolini et al. (2014) [11] compared the use of accelerometers placed on the heel and metatarsal external to the shoe with 2D video analysis to determine foot strike patterns. The authors compared the time between heel and metatarsal accelerations and foot strike angle obtained from the video analysis. The method was determined to be reliable for a wide range of speeds, slopes, and foot strikes. In another application, two IMUs (one accelerometer and one gyroscope) on the top of the shoe were used to determine foot strike angle during running. Researchers found a significant correlation between their determinants of strike angle and sagittal plane angles from a 3D motion camera system [16]. 

The above-mentioned techniques made use of two separate IMUs, either accelerometers, or a combination of an accelerometer and a gyroscope to determine the characteristics of foot strike during running. Furthermore, all of the IMUs used in these studies were mounted on the surface of the shoe. Potential sensor movement with respect to the shoe reduces the ability of the sensor to accurately track the actual foot movement. It has also been shown that the addition of weight on the exterior of the shoe affects the metabolic cost of running [17]. Although current sensors are lightweight, placement at a distance from the center of the foot could potentially affect running economy. Recently, a wireless, skin-mounted, and conformal inertial sensor (BioStampRC, mC10 Inc., Lexington, MA, USA) has been developed with the ability to collect 3D accelerometer and 3D gyroscope data as well as sense electrical activity. The fact that this sensor is soft, thin, and conformable allows it to be placed directly on the foot to give a more accurate representation of foot dynamics. This allows the sensor to be worn inside the shoe during shod running, but also allows for the analysis of foot dynamics during barefoot running. With the BioStampRC’s onboard memory, the potential for data collection in the field is expanded, and the ability to collect data wirelessly allows for movement without restriction.

Therefore, the purpose of this study was to validate the use of the BioStampRC in determining foot strike pattern against traditional video analysis techniques using a high-speed video camera. Specifically we aimed to determine: (i) whether these sensors could accurately detect changes or differences in foot strike angle during running, and (ii) whether the sensors could detect different FSPs (i.e., rearfoot (RF), midfoot (MF), forefoot (FF)) during running. Apart from considering FSP, our first aim was to determine whether the sensor could accurately detect foot strike angle. Since FSP is often determined by cut-off values in foot strike angle [6,18,19], the foot strike angle itself might also be of interest. It is plausible that a runner can significantly change foot strike angle without necessarily changing to a different FSP. Altman and Davis (2012) have suggested that foot strike angle might be a better measure as it is an ‘objective, quantifiable, continuous indication of FSP’ [4]. It was hypothesized that there would be a positive correlation between the sensor determined foot strike angle and that determined from a more traditional method (2D videography). Furthermore, it was hypothesized that the sensor would be able to distinguish between different FSPs, similar to human raters.

## 2. Materials and Methods

### 2.1. Participants

The data from 12 participants were collected and analyzed in this study. The participants were healthy men and women (no injuries in the past three months, no diabetes, cardiovascular, or renal/kidney disease) aged between 18–45 years old. Participants completed the 2015 American College of Sports Medicine Exercise Pre-participation Health Screening Form and provided informed consent approved by the Institutional Review Board (IRB) of the Appalachian State University. The participants were cleared for vigorous activity in order to participate in this study. 

### 2.2. Experimental Protocol

Subjects reported to the Biomechanics Laboratory for one visit lasting about an hour. Height and mass (height: 1.76 ± 0.12 m; mass: 75.2 ± 23.2 kg) were collected and participants were given five minutes to warm up at a self-selected resistance and cadence on a cycle ergometer (Monark Exercise AB, Vansbro, Sweden). Before beginning a trial, participants were asked to stand stationary on a treadmill (Bertec, Columbus, OH, USA) followed by a jump into the air. This jump was later used as an event to synchronize data collection. Participants then began running at a self-selected comfortable running pace on the treadmill. If the participant did not know a comfortable running pace, the speed of the treadmill was modified until a comfortable speed was found. Participants ran for one minute using either a FF strike pattern or a RF strike pattern (randomized). After a one-minute rest, the opposite strike pattern was performed. Finally, after both FF and RF trials were completed, participants were asked to run a third trial using a MF strike pattern. If a subject was unfamiliar or unaccustomed with a specific FSP, the FSP was described similar to the definition by Hasegawa et al. (2007) [6].

### 2.3. Data Collection

Prior to the participant starting the trials, a BioStampRC sensor (mc10, Lexington, MA, USA) was placed on the dorsal side of the right foot underneath the sock and shoe to measure gyroscope and accelerometer data (Figure 1). The sensor (6.6 × 3.4 × 0.45 cm) was lightweight (7 g), soft, and flexible, and contained an inertial measurement unit with a 3-dimensional accelerometer (±16 G) and 3-dimensional gyroscope (±2000°/s). Data were stored in the on-board memory. The sensor was attached with a double-sided sticker pressed firmly onto the skin. Accelerometer and gyroscope data were sampled at 250 Hz. Misthcke et al. (2017) [20] showed that the sampling rates of 200 Hz or above are sufficient for kinematic measures such as the angular velocities and angles employed in this study. 

Foot strike was additionally captured by means of a Sentech high-speed USB3 camera (STC-MBS241U3V, Sentech Technologies America, Inc, Carrollton, TX, USA) using MaxTRAQ 2D Software (Version 2.8.1.1, Innovision Systems Inc, Columbiaville, MI, USA). The camera was set up at the level of the treadmill to enable the whole stride and foot strike to be clearly visible in the frame (Figure 1). The frame rate was set at 120 frames per second with a resolution of 800 × 600.

### 2.4. Data Processing

#### 2.4.1. Data Synchronization

Each trial began with the participant performing a standing jump into the air where both feet left the treadmill. This movement was used to time synchronize the videography data with the IMU sensor data. Two-dimensional (2D) videography data were downloaded and analyzed using Kinovea software (Version 0.8.15, www.kinovea.org). Absolute foot angle was measured over the time of the initial jump (from just before the participant took off for the jump until just after the participant landed). IMU sensor data, specifically gyroscope (angular velocity) data measured about the z-axis (corresponding to sagittal plane axis of the foot-placed sensor), were integrated to give estimated sensor foot angles. Synchronization occurred by means of a MATLAB (Mathworks, Natick, MA, USA) function, which determined which part of the signal (2D angles of jump from videography) best matched a longer data stream (IMU sensor based angles). After the videography and sensor angle data were resampled (using shape-preserving piecewise cubic interpolation) to a common frequency (1000 Hz), this synchronization method shifted the signal (2D videography angle during jump) to best match the data (IMU sensor based angles throughout the complete trial) by calculating the smallest squared Euclidean distance between the signal and data array. This allowed a time match of the different signals with sufficient accuracy to analyze the same foot contacts using the sensor and the videography data. Once the two datasets (videography and IMU data) were synchronized, a time 30 s into the trial was selected as the starting point for foot strike analysis. The first ten right foot strikes after this starting point were further analyzed. 

#### 2.4.2. Foot Strike Determination

Human rater using 2D videography: Ten steps were analyzed by three independent raters using two sets of criteria: Video Foot Strike Angle (FSA_VIDEO_): Using Kinovea software, raters measured the foot angle of each foot strike relative to the treadmill surface. The angle was measured between the top surface of the treadmill belt and the bottom surface of the outsole of the shoe with the vertex at the initial shoe contact point with the treadmill. A positive angle indicated rearfoot and a negative angle indicating forefoot (Figure 1). Final FSA_VIDEO_ was the average across raters.Video Foot Strike Classification (FSC_VIDEO_): Determining an ordinal classification of foot strike pattern based off of Hasegawa et al. (2007) [6], raters were given the following instructions related to whether a footfall was RF, MF, or FF (Figure 1):
Rearfoot strike (RF)—first foot-ground contact with the heel or rear third part of the sole only. Midfoot or forefoot portion had no contact at foot strike.Midfoot strike (MF)—first foot-ground contact with not only the rear third of the sole, but the midfoot or entire part of the sole.Forefoot strike (FF)—first foot-ground contact was the forefoot, or front half of the sole, and the heel did not have any contact at foot strike.

For final analysis, foot strikes that were given the same rating by at least two of three raters were used, which were all of the 360 foot strikes (i.e., no foot strike was given three different classifications of RF, MF, FF) by the three different raters. However, after analyzing the results from the rater classification, we found that only 12% (44 out of 360) of all foot strikes were classified as MF, and that 5 out of 12 participants never performed a MF strike pattern as rated by the raters. The low number of samples reduced the potential accuracy of the classification of MF strike pattern and valid discrimination between RF, MF, and FF. We therefore decided to collapse all of the MF and FF foot strikes into one category: non-rearfoot strikes. The two final classifiers were: rearfoot (RF) and non-rearfoot (NRF).

BiostampRC sensors: All data were downloaded and processed via custom MATLAB code (Mathworks, Inc., Natick, MA). Sagittal plane gyroscope data were integrated to get the foot angle. From the foot angle data, it was possible to determine a stride, which was defined as the period from the maximal foot angle magnitude to the next maximal foot angle magnitude (Figure 2A). Visual observation indicated that the time of the maximal foot angle magnitude coincided with the start of the foot forward swing, and that foot strike occurred somewhere in the middle of this strike period. Foot strike was then determined from the 3D accelerometer data by calculating the peak of the resultant acceleration in each stride period (Shiang et al., 2016) [16] (Figure 2B):Sensor Foot Strike Angle (FSA_SENSOR_): After foot strike was determined, the difference in foot angle at foot strike and the angle when the foot was deemed to be stationary on the ground (lowest mean resultant acceleration over a 50 ms interval after foot strike) was used to calculate the change in foot angle (Figure 2C). A more positive angle indicated a more RF strike pattern:FSA_SENSOR_ = θ_STATIONARY_ − θ_FOOTSTRIKE_(1)Sensor Foot Strike Classification (FSC_SENSOR_): The final rater classification analysis considered only the RF and NRF foot strikes, therefore it was decided to classify FSP using a measure of initial foot strike angular velocity. Sagittal plane gyroscope angular velocity (ω_FS_) over the first 15 ms after foot strike was averaged to get an indication of the direction of angular rotation directly after foot contact (Figure 2D). It was argued that a positive ω_FS_ would indicate a more RF strike pattern, and that a negative ω_FS_ would indicate a more NRF strike pattern. Therefore, FSC_SENSOR_ is a function of the average angular velocity over the first 15 ms after the initial foot strike:FSC_SENSOR_ = *f*(ω_FS(0–15ms)_)(2)

To determine the most appropriate cut-off value to distinguish between RF and NRF using ω_FS(0–15ms)_ data, a Youden‘s index approach was used [21]. Youden’s index can be described as a classification threshold that maximizes the effectiveness of a classifier. This method, commonly used to measure diagnostic effectiveness, is intended to optimize the classifier’s differentiating ability between conditions when equal weight is given to both conditions such as in this case [22]. In our case, the Youden index was defined as the ω_FS(0–15ms)_ cut-off point that maximized the sum of the accurate classifications of RF (true RF classification/all RF classification) and NRF (true NRF classification/all NRF classification). The FSC_SENSOR_ could then determine the classification decision based on the value of ω_FS(0–15ms)_ with respect to the determined cut-off point.

### 2.5. Statistical Analysis

Interrater agreement between the three human raters for foot strike angle was assessed using interclass-correlation (two-way mixed method with absolute agreement) with the quality of agreement assessed using the following cut-offs: <0.50 = poor; 0.50–0.75 = moderate; 0.75–0.90 = good; >0.90 = excellent [23]. Fleiss kappa (κ) was used to determine the interrater agreement with respect to FSP analysis, with the following values to determine quality of agreement: <0.40 = poor; 0.40–0.75 = intermediate to good; and >0.75 = excellent [24]. The relationship between the foot strike angle measurements by the sensor and raters (FSA_SENSOR_ vs FSA_VIDEO_) was compared using a Pearson-product moment correlation. This was analyzed within participants and across all participants using all measured values across all foot strikes. Differences between ω_FS(0–15ms)_ for the RF and NRF foot strikes were analyzed using a *t*-test, with the cut-off point to define the FSC_SENSOR_ from ω_FS(0–15ms)_ data determined using the Youden index. All statistical analyses were done in SPSS, with statistical significance (alpha) set at 0.05.

## 3. Results

The interrater reliability measured showed excellent agreement for foot strike angle with interclass-correlation coefficient = 0.963 (95% confidence interval: 0.954–0.971). FSP analysis also showed excellent agreement with κ = 0.884 (95% confidence interval: 0.838–0.941) when analyzing the three categories (RF/MF/FF), and κ = 0.941 (95% confidence interval: 0.881–1.000) when FSP was collapsed to two categories (RF/NRF). 

There was a very strong and significant correlation between FSA_SENSOR_ and FSA_VIDEO_ across all participants (*r* = 0.868, *p* < 0.001) with the following linear regression relationship: FSA_VIDEO_ = 1.4925·FSA_SENSOR_ + 11.194. Individual correlations within the participants varied between *r* = 0.832 to *r* = 0.994, and were all significant (*p* < 0.001) (Figure 3). 

FSC_VIDEO_ classified 49% (177/360) of foot strikes as RF, 12% (44/360) as MF, and 39% (139/360) as FF. Collapsing MF and FF into one category resulted in 51% (183/360) of NRF foot strikes. There was a significant difference between the ω_FS(0–15ms)_ associated with foot strikes that were classified as RF and those classified as NRF by raters using video (FSC_VIDEO_) (RF: 160 ± 137°/s; NRF: −256 ± 149°/s, *t* = 27.591, *p* < 0.001) (Figure 4). To establish a sensor-based classification (FSC_SENSOR_), the Youden analysis suggested a cut-off value for ω_FS(0–15ms)_ of 2.8°/s. This implies that when ω_FS(0–15ms)_ > 2.8°/s, a foot strike is classified as RF and when ω_FS(0–15ms)_ ≤ 2.8°/s, a foot strike is classified as NRF for FSC_SENSOR_ classification. For the population in this study, this allows for a 92.5% accuracy in classification. Since this value of 2.8°/s is based off the ranked values (in the original set of the data that ranges from −548.0°/s to 503.4°/s), it is evident that a cut-off of 0°/s, which is expected to differentiate between different directions of foot rotation, should also be considered. We therefore compared these values: 2.8°/s (54.0 percentile score) to 0.0°/s (53.7 percentile score) as the cut-off. Changing from 2.8°/s to 0°/s had a minimal change in the accuracy of classification from 92.5% (333/360) to 92.2% (332/360).

## 4. Discussion

As hypothesized, our results indicated that there was a strong and significant correlation between the foot strike angle measured by the BiostampRC sensor (FSA_SENSOR_) and the values measured using 2D videography (FSA_VIDEO_). Similar studies correlating IMU sensor based indices of foot strike angle and foot strike angles measured through videography have found coefficients between *r* = 0.74 [25] to *r* = 0.98 [16]. Our results (*r* = 0.87, *p* < 0.001) are comparable to the results from these studies. The linear regression equation results do show a bias (offset of 11.194°) between the two measures, suggesting that FSA_SENSOR_ consistently underestimated FSA_VIDEO_. This bias should be considered when using the suggested method. Similar studies have incorporated multiple sensors to determine the foot strike angle, with some incorporating two accelerometers [11,25] and also a separate accelerometer and gyroscope [16]. Only Falbriard and colleagues (2017) [26] used a similar single sensor (affixed to the upper part of shoe) approach by integrating gyroscope data to determine foot strike angle, with results (*r* = 0.89) very similar to ours. Our correlation results are promising and suggest a link between sensor based and human rater based foot strike angle estimates. However, since a large bias did exist between the methods, the results should be considered with caution. Any potential biases should be taken into account when using the suggested framework. 

We additionally investigated the idea of using a cut-off value determined by angular velocity (ω_FS(0–15ms)_) to classify different foot strike patterns. We decided to focus only on the classification between RF and NRF, although the aim and our initial hypothesis were to be able to distinguish between three different FSPs (RF, MF, FF). This change to a binary classification, where foot strike could be classified as either RF or NRF (FF or MF), was done due to a limited number of participants using a MF strike during running (as determined by three independent human raters). Creating a classifier cut-off based off the Youden method gave a value of 2.8°/s as the best cut-off for this participant group. This result was not distinctly different from a value of 0°/s, which is ultimately an intuitive cut-off point for differentiating between foot strikes; a positive value can indicate RF and a negative value can indicate NRF. Although this result appears to be trivial, to our knowledge this has not been evaluated previously. Previous work have used different methods to classify FSP into different categories, be it RF/MF/FF or RF/NRF. Giandolini et al. (2014) [11] employed a method that determined time differences between acceleration peaks of the heel and metatarsal to infer foot strike classification. This method was purely based on accelerometer data. Our results using one sensor employed both accelerometer data (to determine foot strike) and gyroscope data (to determine angular velocity). Shiang et al. (2016) [16] used a foot strike index (based off integrated gyroscope data) to determine a cut-off point between RF and FF strike. Their method considered angle changes that occurred from before the foot strike until a certain time after foot strike. It is unclear how the effect of foot strike angle changes before foot contact determined foot strike classification, as used by Shiang and colleagues (2016) [16]. Our method used average angular velocity only after foot strike occurred. 

Difficulty in differentiating MF and FF foot strikes has been previously acknowledged [25]. Using two uniaxial accelerometers and a method similar to Ginadolini et al. (2014) [11], Gaudel and colleagues [25] mentioned that when compared to visual observations, similar to this study, misclassification for MF strike could occur. Further research is necessary in the potential use of a single IMU to accurately differentiate between the three FSPs. However, it must be acknowledged that FSP does not necessarily occur as discrete values [4], which might be a reason why it is difficult to have accurate classification into three separate patterns. Our study had other specific limitations. Our initial aim was to have participants run with three different FSPs in order to obtain a wide range of foot strike angles to measure. The peak rating of the accelerometers used in this study was ±16 g. It has been shown that foot peak accelerations could be larger than the 16 g operating range [27], which could have potentially affected the correct determination of the time of initial foot strike. Foot strike angles determined using sensors (FSA_SENSOR_) underestimated the angles as measured by human raters (FSA_VIDEO_). Interrater reliability values appear to indicate that the human raters were consistent in determining foot strike angle and the use of Kinovea software to measure angles has been shown to be reliable and valid [28,29]. Regarding sensor data, one possible reason for this offset is the potential underestimation of time over which the angular velocity data were integrated to determine foot strike angle. Initial foot contact was determined at the time point when maximum acceleration of the foot occurred, as suggested by Shiang et al. (2016) [16]. Others have employed different techniques, which might estimate an earlier time point of actual foot contact, i.e., before maximal foot acceleration occurs [30,31]. An earlier foot contact estimate would increase the time of integration and lead to larger, potentially more accurate, foot strike angles being estimated. Similarly, inaccurate estimates of the time at which the foot reaches the stationary position could also cause errors in angle calculations. The accurate determination of these parameters, which is dependent on the signals used (e.g., single axis or resultant acceleration), the site of the sensor attachments, etc., should be critically considered in future studies.

To our knowledge, previous attempts to quantify foot dynamics using IMUs have used sensors attached to the shoe of the participant. The sensor used in this study had the ability to be directly attached to the foot surface. Due to the sensor being thin and conformable, a regular sock and shoe could be worn over the foot and sensor. Although in this study we did not test the sensors during barefoot running, it is equally possible to wear the sensors without shoes, which is more difficult for the larger, rigid sensors used in previous studies. In addition, it has been suggested that it is important to reduce the external oscillation of sensors worn on the shoe surface, and care should be taken in how these sensors are attached [11]. The sensor in this study was directly attached to the foot and minimal movement or oscillation was expected, potentially decreasing errors from movement artifacts and thereby making a more direct measure of the foot dynamics achievable. 

The sensors used in this study (BiostampRC, mc10 Inc., Lexington, MA, USA) have also been previously used for other applications. For example, McGinnis and colleagues (2016) [32] validated the use of the sensors to accurately measure knee joint angles during functional knee assessment activities by using two sensors (attached to the thigh and shank). Their results showed that the sensors were able to track differences in knee angle to within 1% of standard goniometer values and the knee range of motion measures agreed well with the standard goniometer measures. Others have shown that the sensors attached to the shank could be used to differentiate gait characteristics (e.g., step time, swing time, step number) between people with multiple sclerosis of different disability levels [13]. It is foreseeable that sensors like these could possibly be used in a variety of settings, performance or clinical, with both inertial measures and electrical activity measures (not tested here) in order to perform more comprehensive motion analysis outside the traditional laboratory.

## 5. Conclusions

The use of thin, conformable sensors directly attached to human skin is an attractive option to accurately measure human motion outside of a laboratory setting. The main finding of this study showed that using a fairly simple method of angular velocity estimates and an intuitive 0°/s as an angular velocity cut-off value allowed the sensor to distinguish between a rearfoot and non-rearfoot strike pattern. Furthermore, there was a strong and significant correlation between the foot strike angle measured using the BiostampRC sensor and the values measured using 2D videography. Exploring and validating these sensors should be of interest as a means to improve accurate measurements of human motion in non-laboratory settings. 

## Figures and Tables

**Figure 1 sports-07-00184-f001:**
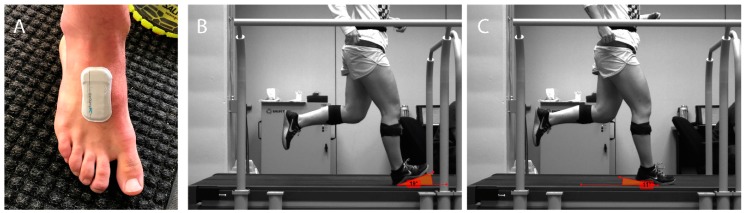
(**A**) BiostampRC sensor attached to the dorsal surface of the foot to determine sensor based foot strike angle (FSA_SENSOR_) and foot strike classification (FSC_SENSOR_). (**B**,**C**) View from camera showing the participant running on treadmill. This view was used by human raters to determine the foot strike angle from the video (FSA_VIDEO_) and the foot strike classification from video (FSC_VIDEO_). The image on the left (**B**) shows an example of FSA_VIDEO_ = 19° and FSC_VIDEO_ = rearfoot (RF). On the right (**C**) is an example of FSA_VIDEO_ = −11° and FSC_VIDEO_ = forefoot (FF).

**Figure 2 sports-07-00184-f002:**
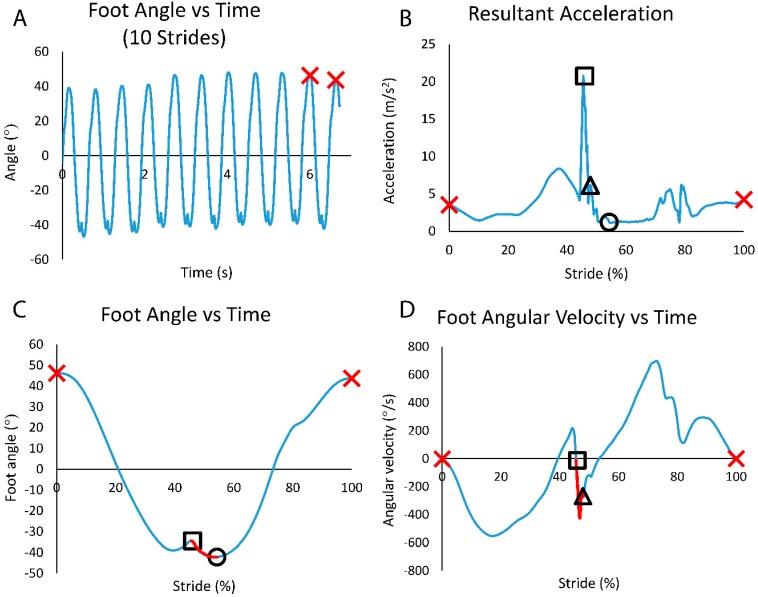
Example data of a participant running with a non-rearfoot strike pattern. (**A**) Foot angle vs. time over the 10 strides to be analyzed (x’s show the start and end of one stride). (**B**) Resultant acceleration within the single stride period showing peak acceleration (foot strike) (□); the time 15 ms after peak acceleration (∆); and the time of the start of the lowest 50 ms average acceleration (indicating stationary foot) (○). (**C**). Foot angle over one stride. Change in foot angle (FSA_SENSOR_) determined from the difference between foot stationary angle (○) and initial foot strike angle (□). (**D**) Foot angular velocity over one stride. Average angular velocity, ω_FS(0–15ms)_ calculated in intervals from foot contact (□) to a point 15 ms later (∆).

**Figure 3 sports-07-00184-f003:**
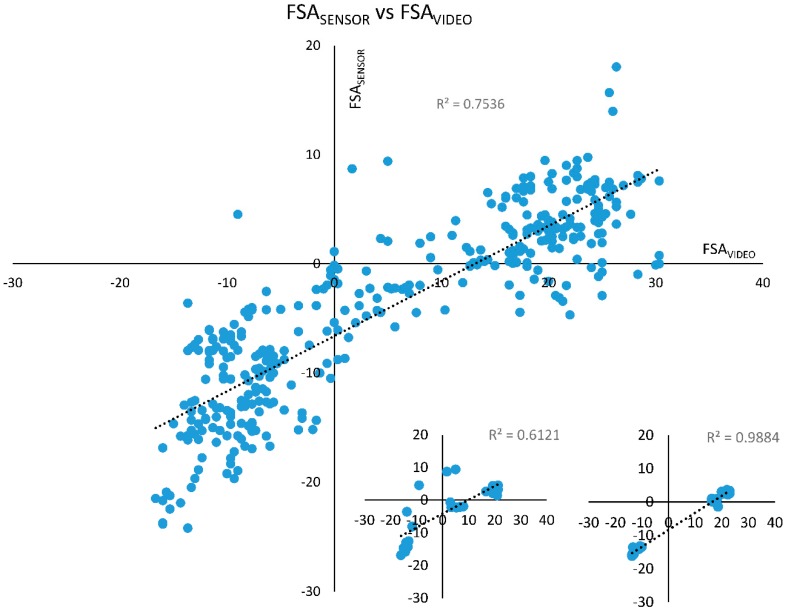
FSA_SENSOR_ (y-axis) plotted against FSA_VIDEO_ (x-axis). There was a very strong and significant correlation between FSA_SENSOR_ and FSA_VIDEO_ (*r* = 0.868, *p* < 001). Inserts: Within participant data showed significant correlations with *r* = 0.832 (*p* < 001) for the worst case (left insert) and *r* = 0.994 (*p* < 001) for the best case (right insert).

**Figure 4 sports-07-00184-f004:**
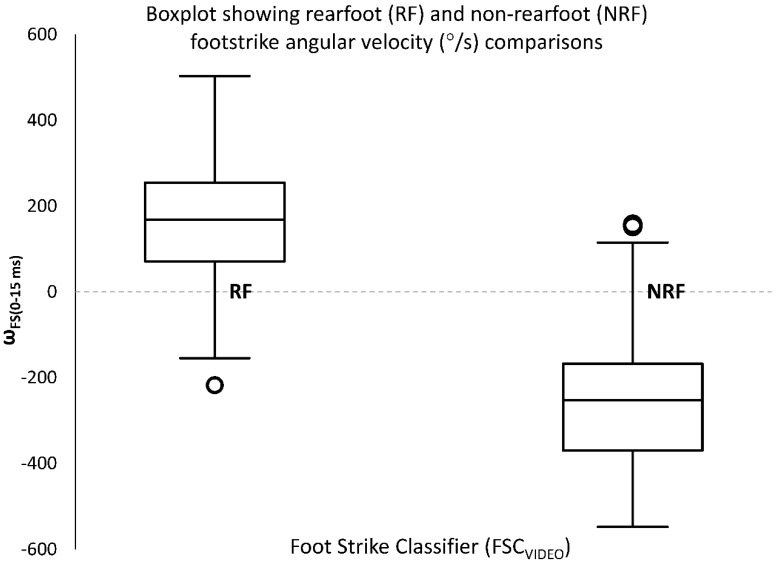
Boxplot showing differences in average angular velocity at foot strike (ω_FS(0–15ms)_) for different foot strike classification made by human raters using 2D videography data (FSC_VIDEO_). There was a significant difference between ω_FS(0–15ms)_ for RF and NRF (*t* = 27.591, *p* < 001).

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
