# Peer review of "Using A Soft Conformable Foot Sensor to Measure Changes in Foot Strike Angle During Running"

_sports, 2019, doi:10.3390/sports7080184_

Reviewer 1 Report

The main aim of this study was to validate an IMU sensor placed directly on the dorsal foot surface to determine foot strike angle and pattern. The results of this study show that a directly to the human skin attached sensor can be used to classify the runners foot strike pattern (rearfoot or non-rearfoot) and that there is a high correlation between the calculated footstrike angle with an IMU and the by three observers rated foot strike angle.

The premise behind this work is interesting to the general running research community. Overall, the manuscript is well written. There are few areas of concern, which the authors should address to improve this manuscript.

Abstract

Line 14: Are IMUs really placed on lower legs to measure foot strike pattern? Please write “on the heel cap or on the instep of the shoe”

Line 18: “… 2D video analysis techniques.” Please add “2D”

Line 21: What is about the Bias that was mentioned in line 249?

Introduction

Line 27: I agree with the importance of determining the foot strike pattern RF – MF – FF (this was shown with your added literature). But why is the accuracy of the foot strike angle important? Please add literature.

Line 29: Are these “lab settings” treadmills only or also indoor tracks?

Line 41: What means “on the toe of the shoe”? Which toe?

Line 54: “… were mounted on the surface of the shoe.” Is that a problem? If so, please describe in more detail.

Line 67: Please define “RF”, “MF”, and “FF”

Line 67: What means “a non-zero correlation”? I think that r = 0 is not possible.

Material and Methods

Line 82: Why was the warm up on a cycle ergometer and not on the treadmill?

Line 85: Were all participants experienced treadmill runners? If not, was there a familiarization time for these runners?

Line 91: Please at a point at the end Hasegawa et al

Line 100: Is a sampling rate of 250 Hz sufficient to measure foot strike ankle accurately with an IMU during running? Can you add literature?

Line 123: Was the “resampling” a linear interpolation or a spline fitting? Please add information

Line 136: Which part of the shoe means “bottom of shoe”? Was there a characteristic point on the bottom of the shoe or was it chosen individually in relation to the footwear of the runners?

Line 157: “MATLAB” or ”Matlab” (Line 121). Please be consistent

Line 158: How was the drift handled after signal integration?

Line 160: Please add (Figure 2A)

Line 162: Measures the used foot strike detection method accurately the initial foot ground contact? Especially when investigating foot strike angle at this special moment, the foot strike should be detected as accurately as possible. Our experience according the foot strike detection is that the foot strike is before the maximum acceleration occurs, at the point when the high acceleration starts.

Line 164: Please add (Figure 2B)

Line 168: Please add (Figure 2C)

Line 174: Please add (Figure 2D)

Line 179: Please describe in a few words the Youden’s index

Figure 2: Please add the information that this is a dataset off a non-rearfoot runner

Line 202: “This was analyzed within participant and across all participants.” How was the calculation for “all participants” – mean or median values across all foot strikes? Please add this information

Line 248: What could be the reason of the underestimated foot strike angle of the FSAsensor? Maybe the accuracy of the detection of the initial ground contact, the drift error in the integrated gyroscope signal or the inaccurate calculation of the stationary phase? Maybe an inaccurate tracking in the kinovea program? Can anything be excluded? Please describe. This is an important information for further studies with this sensor.

Line 254: I do not agree with your statement. Figure 3 shows that a lot of IMU calculated foot strike angles were underestimated. In this example, a high correlation does not mean that with an IMU it is possible to calculate accurate foot strike angles like raters can. In many cases raters rated positive angles, but the IMU calculated negative values. This leads to completely different conclusions when interpreting results. Please revise this statement.

Line 264: Was done a test of significance? If not, please change to “distinctly different”. Otherwise, please add the test in the methods.

Line 278: Please add a point after Ginadolini et al

Reviewer 2 Report

In relation to this article that has been sent to us for appreciation we have to mention that it presents very small gaps. Thus it is conditioned to only a few small changes.

The article is well written, with the appropriate methodology and an updated bibliography.

In the Introduction, the line 67 must be changed:

(i.e. RF, MF, FF) ??? write in full please!
